# Occupational exposure to styrene and acute health effects among fiberglass-reinforced plastic workers: An integrated environmental and biological monitoring study

Oh-Hyun Kwon[1], Ki-Youn Kim[1,2]*

1 Graduate School of Safety Engineering, Seoul National University of Science and Technology (SeoulTech), Seoul, Republic of Korea, 2 Department of Safety Engineering, Seoul National University of Science and Technology (SeoulTech), Seoul, Republic of Korea

* kky5@seoultech.ac.kr

## Abstract

Styrene remains a major hazard in fiberglass-reinforced plastic (FRP) manufacturing. The current 10 ppm 8-h TLV-TWA is half the former limit, and the movement by several European states toward comparable or lower OELs highlights the need for fresh exposure–response data. In this study, eighty-five Korean FRP workers were monitored cross-sectionally. Full-shift breathing-zone styrene was measured by GC-FID; post-shift urine was analysed for mandelic acid (MA) and phenyl-glyoxylic acid (PGA), and dermal uptake was estimated with fluorescent tracers. Neuro-irritative symptoms were assessed by questionnaire and clinically verified in a subset. The results showed that median styrene levels were 18.65 ppm (spray-up), 12.42 ppm (hand lay-up) and 6.37 ppm (closed-mold). Urinary MA and PGA correlated with air levels (r = 0.78, 0.77). Dermal styrene load showed a moderate correlation with urinary MA (r = 0.42, p < 0.001). Symptom prevalence rose from 19% to 71% across exposure quartiles (adjusted OR = 5.6). A biomarker-based model using urinary mandelic acid (MA) with covariates (age, ventilation) showed strong apparent discrimination (AUC = 0.93). We propose 0.38 mg/g creatinine (MA) as a candidate operational ("early-warning") threshold, pending external validation. In conclusion, integrated air, biological and dermal metrics reveal dose-dependent acute effects at or below 10 ppm. In this cross-sectional analysis, higher styrene exposure was associated with increased acute symptoms at or below ~10 ppm; these associations warrant confirmation in longitudinal studies with repeated biomonitoring. We present 0.38 mg/g creatinine (MA) as a candidate operational ("early-warning")threshold to flag workers for closer evaluation; external validation is needed, and engineering controls remain the primary means of risk reduction.

**Data availability statement:** All relevant data files are available from the Zenodo repository (DOI: 10.5281/zenodo.17248606).

**Funding:** This study was supported by the Research Program funded by SeoulTech (Seoul National University of Science and Technology).

**Competing interests:** No authors have competing interests.

# 1. Introduction

Styrene ($C_8H_8$) is a pivotal industrial monomer predominantly used in the synthesis of polystyrene plastics and unsaturated polyester resins [1,2]. The global styrene market, estimated at approximately $35 billion in 2023, is experiencing growth driven by the increasing demand in the fiberglass-reinforced plastic (FRP) sector, notably in boat manufacturing, automotive components, and construction materials [3,4]. In South Korea, the FRP industry contributes over $2 billion annually to the national economy [5]. The production of FRP components typically employs open-mold processes, such as hand lay-up and spray-up techniques, wherein workers manually apply styrene-containing resins to the molds [6,7]. Although these processes are cost-effective and versatile, they pose a significant potential for occupational exposure due to the high vapor pressure of styrene (0.67 kPa at 25°C) and the open nature of the work environment [8]. In contrast to closed-mold processes that contain emissions, open-mold operations can result in airborne styrene concentrations that frequently exceed the established occupational exposure limits [9,10].

Despite extensive research on styrene toxicity over the decades, substantial knowledge gaps persist regarding exposure-response relationships for acute health effects in real-world occupational settings. Most prior studies have focused on chronic health outcomes or have been conducted in controlled laboratory environments that may not accurately reflect actual workplace conditions [11,12]. Moreover, few studies have integrated multiple exposure assessment methodologies, including environmental monitoring, biological markers, and quantified dermal exposure, to provide a comprehensive understanding of the total internal dose of these chemicals [13,14].

In 2018 the American Conference of Governmental Industrial Hygienists (ACGIH) proposed lowering the styrene 8-h Threshold-Limit-Value Time-Weighted-Average (TLV-TWA) from 20 ppm to 10 ppm; the cut was formally adopted in the 2020 TLV list (STEL unchanged at 20 ppm). Several European countries have since moved toward comparable or stricter limits: Sweden and Spain now apply a 10 ppm national OEL, Norway has adopted 5 ppm, while others, such as Germany and the Netherlands, still retain 20 ppm. These tightening or impending benchmarks underscore the need for contemporary exposure–response data that reflect current workplace conditions.

Large-scale- environmental studies hint at short-term- neuro-irritative consequences: among 21,962 Gulf Coast residents, those in the highest quartile of ambient styrene showed a 12–17% excess prevalence of neurologic symptoms compared with the lowest quartile. However, occupational investigations that simultaneously capture breathing zone- air, urinary metabolites, and quantified dermal uptake and relate them to real-time- health outcomes remain scarce, particularly in Asia. This gap is striking, given South Korea's sizable and growing FRP-manufacturing workforce.

Therefore, we (i) characterized the total internal styrene dose by measuring concurrent air, urinary, and dermal metrics in Korean FRP workers; (ii) modelled dose-dependent- relationships with self-reported- and clinically verified neuro-irritative symptoms; and (iii) derived a biomarker-based- action level of 0.38 mg/g creatinine to guide workplace interventions. We hypothesized that adding dermal

fluorescence intensity to urinary mandelic acid levels would outperform airborne metrics alone in symptom prediction. The following sections describe our study design, present exposure and health findings, and discuss their implications for occupational hygiene and regulatory policies.

## 2. Methods

### 2.1. Study design and setting

This cross-sectional study was conducted at a prominent FRP boat manufacturing facility in Incheon, South Korea, from March to September 2023. The facility employs approximately 150 workers and specializes in producing recreational boats, commercial vessels, and marine components using various FRP-manufacturing processes. The study protocol was approved by the Institutional Review Board of Seoul National University of Science and Technology (IRB No. 2023-0036-01, approved March 6, 2023) and was conducted in accordance with the Declaration of Helsinki [15]. All participants provided written informed consent prior to enrollment.

### 2.2. Study population

The eligibility criteria for participation included direct involvement in resin handling for a minimum of 2 h per shift or 8 h per week, employment in the current position for at least 6 months, and willingness to participate in all the study procedures. The exclusion criteria were pregnancy, recent use of medications known to interfere with styrene metabolism, and acute illness at the time of the assessment. Of the 92 eligible workers, 85 (92.4%) consented to participate and completed all the study procedures.

### 2.3. Exposure assessment

**2.3.1. Airborne styrene.** Monitoring Personal breathing-zone air samples were collected using calibrated low-flow sampling pumps (SKC Inc., Eighty Four, PA, USA) equipped with activated charcoal sorbent tubes (SKC 226−01). Sampling was conducted over the entire work shift (8 h) at a flow rate of 0.2 L/min. Gas chromatographic analysis was performed using an Agilent 7890A GC system equipped with a flame ionization detector (FID) in accordance with the National Institute for Occupational Safety and Health Method 1501 [16]. The method detection limit was 0.03 mg/m³, and the desorption efficiency was 95.2 ± 2.8%, exceeding the acceptance criterion of ≥90%. Additional details of the GC-FID analytical method are provided in the Supplementary Methods (Section S1.1 in S1 File).

**2.3.2. Biological monitoring.** Post-shift urine samples were collected on two consecutive days and analyzed for mandelic acid (MA) and phenylglyoxylic acid (PGA) using high-performance liquid chromatography with UV detection (HPLC-UV) following a validated acid hydrolysis procedure employing 6 M HCl at 100°C for 16 h [17,18]. The acid hydrolysis method was validated against enzymatic hydrolysis and demonstrated excellent agreement (MA: 98 ± 5%; PGA: 96 ± 6%). The complete HPLC-UV analysis protocol and quality control measures are available in Supplementary Section S1.2 in S1 File. Urinary creatinine was measured using the Jaffe reaction to adjust for urine dilution [19].

**2.3.3. Dermal exposure assessment.** To quantify dermal styrene uptake, we employed a fluorescent-tracer technique adapted from validated protocols. A non-toxic optical brightener (Uvitex OB, 0.05 wt%) was premixed with the styrene resin each sampling day. At shift-end, workers' hands and forearms were photographed under 365 nm UV light (Canon EOS R6, 420 nm long-pass filter). Fluorescent tracer–positive skin area (cm²) was quantified using ImageJ (NIH, USA) and converted to mass-based dermal styrene load (µg cm$^{-2}$) via laboratory calibration (R² = 0.98). Duplicate blanks confirmed a method detection limit of 5 µg cm$^{-2}$.

In addition to the primary models, we evaluated an alternative specification including dermal exposure; diagnostics and results are provided in Tables S4–S5 in S1 File.

## 2.4. Health outcome

Acute neuro-irritative symptoms were assessed using a validated questionnaire adapted from previous occupational health studies [20,21]. The questionnaire evaluated the presence and severity of eye, throat, headache, dizziness, and nasal irritation experienced during the current work shift. Symptoms were considered present if they occurred during working hours and were not attributable to other factors. A subset of participants (n = 21) underwent objective clinical measurements, including tear film break-up time (TFBUT) assessment(see Supplementary Methods S1.4 in S1 File) [22].

## 2.5. Statistical analysis

Statistical analyses were performed using R (version 4.3.0). Continuous variables were assessed for normality using the Shapiro-Wilk test and are reported as median (interquartile range) for data not following a normal distribution or as mean ± standard deviation for data that were normally distributed. Group comparisons were performed using the Kruskal-Wallis test for continuous variables and the chi-square test for categorical variables. The correlations among the exposure variables were evaluated using Spearman's rank correlation coefficients. Spearman correlations between dermal styrene load and urinary MA/PGA were also computed, and dermal load was entered into multivariable regression models alongside airborne exposure. Logistic regression models were employed to investigate the exposure-response relationships for acute symptoms. Exposure quartiles were established based on the distribution of the TWA of styrene concentrations. Multivariable models were adjusted for potential confounders, including age, smoking status, and the ventilation status. The model fit was evaluated using the Hosmer-Lemeshow goodness-of-fit test. Statistical significance was set at $p < 0.05$.

Smoking status was ascertained by questionnaire and coded as current/former/never (primary models: current vs. non-current). In a non-smoker–only sensitivity analysis, effect directions and magnitudes were materially unchanged (Table S3 in S1 File).

Airborne and dermal metrics were correlated, and dermal uptake can lie on the pathway to urinary metabolites; to avoid multicollinearity and potential over-adjustment, we prioritized parsimonious models. Nevertheless, we report correlation/ VIF diagnostics (Table S4 in S1 File) and an alternative model including dermal exposure (Table S5 in S1 File).

Sensitivity analyses additionally assessed link functions (logit/probit/clog-log), restricted cubic splines (3–5 knots), cluster-robust SEs/mixed-effects, and outlier influence (top/bottom 1% winsorizing). For transport validation, we used temporal hold-out and leave-one-group-out by task/site, summarizing AUC, Brier score, and calibration intercept/slope (Tables S2–S5; Fig S1 in S1 File).

## 3. Results

### 3.1. Study population characteristics

The study ultimately included 85 male employees, with a median age of 41 years (IQR: 36–46 years; range: 28–55 years). These individuals had worked at the facility for a median duration of 15 years (IQR: 8.5–22 years) and had been in their current roles for a median of 4 years (IQR: 2–6.5 years). Among the participants, 34.1% (29 out of 85) were current smokers, and an additional 21.2% had previously smoked. A significant proportion of workers (63.5%) had completed at least a high school education. (Table 1) provides a detailed overview of demographic and occupational characteristics.

### 3.2. Exposure assessment results

**3.2.1. Airborne styrene concentrations.** There was a notable variation in the personal TWA styrene levels across the different task categories (Kruskal-Wallis $p < 0.001$). The highest median concentration was observed during spray-up operations at 18.3 ppm (IQR: 14.2–22.1 ppm), followed by hand lay-up tasks at 11.2 ppm (IQR: 8.9–14.6 ppm) and closed-mold operations at 7.1 ppm (IQR: 5.2–9.8 ppm).

**Table 1. Demographic and Work Characteristics of Study Participants (n = 85).**

| Variable | Overall (n = 85) | Closed-mold (n = 28) | Hand lay-up (n = 28) | Spray-up (n = 29) | p-value |
|---|---|---|---|---|---|
| **Demographics** | | | | | |
| Age (years)[a] | 41.0 (36.0-46.0) | 40.0 (35.5-45.5) | 41.5 (36.5-46.5) | 41.0 (36.0-46.0) | 0.678 |
| BMI (kg/m²) | 24.8 ± 2.7 | 24.5 ± 2.4 | 25.0 ± 2.8 | 24.9 ± 2.9 | 0.734 |
| **Education level** | | | | | 0.456 |
| ≤ Middle school | 31 (36.5) | 9 (32.1) | 12 (42.9) | 10 (34.5) | |
| **≥High school** | 54 (63.5) | 19 (67.9) | 16 (57.1) | 19 (65.5) | |
| **Smoking Status** | | | | | 0.623 |
| Current smokers | 29 (34.1) | 8 (28.6) | 11 (39.3) | 10 (34.5) | |
| Former smokers | 18 (21.2) | 7 (25.0) | 5 (17.9) | 6 (20.7) | |
| Never smokers | 38 (44.7) | 13 (46.4) | 12 (42.9) | 13 (44.8) | |
| **Employment History** | | | | | |
| Years employed[a] | 15.0 (8.5-22.0) | 16.0 (9.0-23.0) | 14.5 (8.0-21.5) | 15.0 (8.5-22.0) | 0.756 |
| Years in current job[a] | 4.0 (2.0-6.5) | 3.5 (2.0-6.0) | 4.0 (2.0-6.5) | 4.5 (2.5-7.0) | 0.723 |
| **Ventilation ON** | 49 (57.6) | 22 (78.6) | 15 (53.6) | 12 (41.4) | <0.001 |

[a]Data are presented as median (interquartile range). Continuous variables were compared using the Kruskal-Wallis test, and categorical variables were compared using the chi-square test.

A majority of the workers (61.2%) had time-weighted average styrene concentrations above the 10 ppm occupational exposure limit now adopted by the ACGIH (8-h TLV-TWA, formalized in 2020 after the 2018 proposal) and already enforced in several European countries such as Sweden and Spain [23,24]. Fig 1 illustrates the distribution of airborne styrene concentrations by task category.

**3.2.2. Urinary biomarker concentrations.** Significant differences in urinary MA concentrations were observed across the different task categories (Kruskal-Wallis, p < 0.001). The median concentrations were 0.412 mg/g creatinine for spray-up workers, 0.342 mg/g creatinine for hand lay-up workers, and 0.278 mg/g creatinine for closed-mold workers. Similarly, the PGA concentrations showed a comparable trend, with median values of 0.089, 0.076, and 0.062 mg/g creatinine. In comparison to the ACGIH BEI of 0.400 mg/g creatinine for combined MA and PGA, 47.1% of all workers surpassed this limit, including 72.4% of spray-up workers, 50.0% of hand lay-up workers, and 17.9% of closed-mold workers [25]. Fig 2 illustrates the distribution of urinary biomarker concentrations according to the task category. (Table 2) presents the detailed exposure assessment results categorized by task. Additional exposure assessment by work area is shown in Supplementary (Table S1 in S1 File).

**3.2.3. Dermal exposure.** Workers engaged in spray-up tasks exhibited the highest dermal exposure metrics, with a mean fluorescent tracer-positive skin area of 15.2 ± 4.8 cm² and a corresponding dermal styrene load of 220 ± 65 µg cm⁻². The median dermal styrene load was 212 µg cm⁻² (IQR: 170–270).

In contrast, hand lay-up and closed-mold workers had significantly lower exposure, with mean dermal loads of 130 ± 44 and 40 ± 17 µg cm⁻², respectively.

To standardize inter-task comparisons, a dermal exposure index (DEI) was calculated, setting spray-up as the reference (DEI = 1.00). The relative DEIs were 0.59 ± 0.07 for hand lay-up and 0.18 ± 0.05 for closed-mold operations.

These findings highlight a substantial gradient in dermal exposure intensity aligned with task characteristics and proximity to resin application(Table 3).

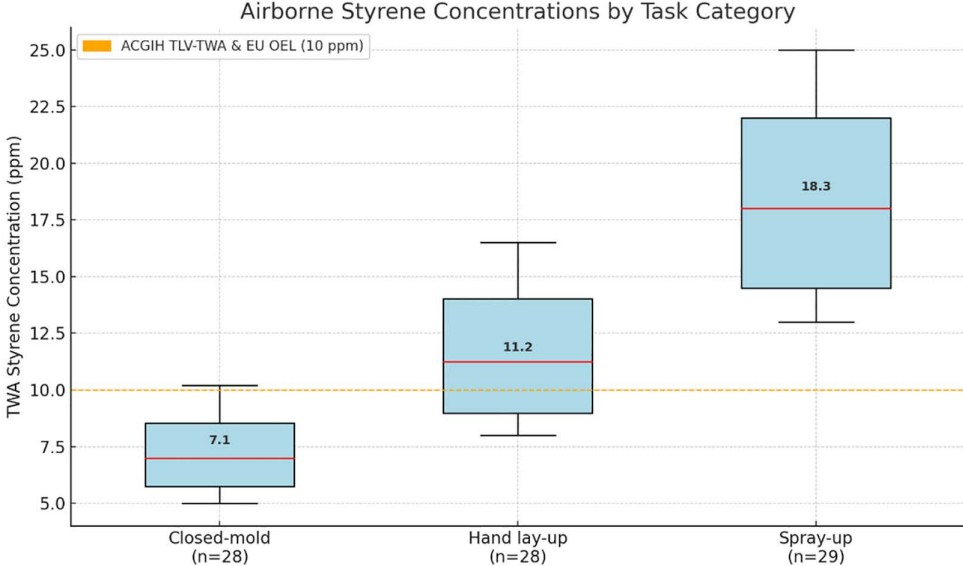

**Fig 1. Airborne Styrene Levels by Task Type.** The box-and-whisker diagrams illustrate the range of time-weighted average (TWA) styrene levels for the three tasks. The spray-up category exhibited the highest median exposure level at 18.3 ppm, followed by hand lay-up at 11.2 ppm and closed-mold operations at 7.1 ppm. Horizontal lines serve as markers for the important occupational exposure limits.

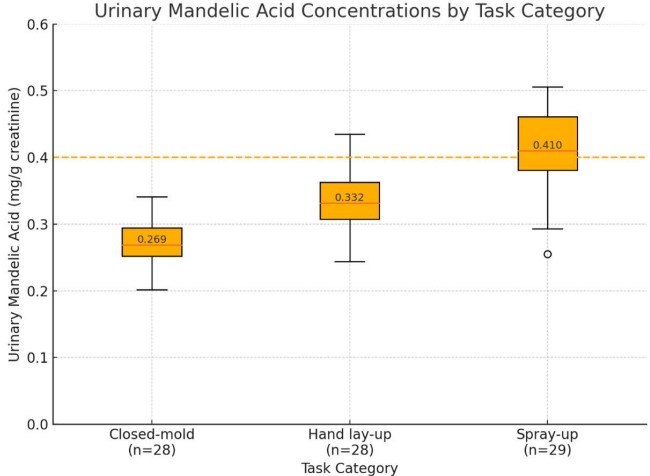

**Fig 2. Urinary Mandelic Acid Levels According to Task Type.** Urinary mandelic acid (mg/g creatinine) by task—Closed-mold (n=28), Hand lay-up (n=28), Spray-up (n=29). Boxes show IQR and medians (0.269, 0.332, 0.410 mg/g creatinine); whiskers=1.5×IQR, circles=outliers. Dashed line marks the ACGIH BEI (0.400 mg/g creatinine).

### 3.3. Exposure-biomarker relationships

Strong positive correlations were observed between the levels of styrene in the air and the metabolite levels in urine. The correlation coefficient between TWA styrene and urinary MA was $r = 0.78$ ($p < 0.001$), and for PGA, it was $r = 0.77$ ($p < 0.001$). The correlation between MA and PGA was even more pronounced, with a coefficient of $r = 0.78$ (95% CI: 0.67–0.86; $p < 0.001$). Dermal styrene load showed a moderate correlation with urinary MA ($r = 0.42$, $p < 0.001$). Linear

**Table 2. Exposure Assessment Results by Task Category.**

| Task Category | N | Styrene (ppm) | MA (mg/g creatinine) | PGA (mg/g creatinine) | Symptoms (%) |
|---|---|---|---|---|---|
| **Closed-mold** | 28 | | | | |
| Median (IQR) | | 7.1 (5.2-9.8) | 0.278 (0.201-0.342) | 0.062 (0.045−0.078) | 21.4 |
| Range | | 3.2-15.6 | 0.145-0.456 | 0.028−0.098 | |
| **Hand lay-up** | 28 | | | | |
| Median (IQR) | | 11.2 (8.9-14.6) | 0.342 (0.268-0.425) | 0.076 (0.058−0.094) | 39.3 |
| Range | | 6.8-19.8 | 0.198-0.578 | 0.042-0.125 | |
| **Spray-up** | 29 | | | | |
| Median (IQR) | | 18.3 (14.2-22.1) | 0.412 (0.345-0.498) | 0.089 (0.072-0.108) | 65.5 |
| Range | | 9.5-28.7 | 0.256-0.687 | 0.054-0.142 | |
| **Overall** | 85 | | | | |
| Median (IQR) | | 12.8 (8.9-18.3) | 0.342 (0.268-0.425) | 0.076 (0.058−0.094) | 41.1 |

MA = mandelic acid; PGA = phenylglyoxylic acid; IQR = interquartile range. p < 0.001 for all comparisons between task categories (Kruskal-Wallis test).

**Table 3. Dermal styrene exposure metrics by task category (n = 85).**

| Task category (n) | Fluorescent tracer–positive skin area (cm²) | | Dermal styrene load (µg cm$^{-2}$) | | Dermal exposure index |
|---|---|---|---|---|---|
| | Mean ± SD | Median (IQR) | Mean ± SD | Median (IQR) | Mean ± SD |
| Spray-up (29) | 15.2 ± 4.8 | 14.5 (11.0–18.5) | 220 ± 65 | 212 (170–270) | 1.00 (ref) |
| Hand lay-up (28) | 8.9 ± 3.1 | 8.4 (6.5–10.5) | 130 ± 44 | 125 (100–160) | 0.59 ± 0.07 |
| Closed-mold (28) | 3.1 ± 1.2 | 3.0 (2.2–3.8) | 40 ± 17 | 38 (28–50) | 0.18 ± 0.05 |
| **Total (85)** | 9.4 ± 6.0 | 8.2 (4.2–13.7) | 140 ± 90 | 120 (60–200) | 0.59 ± 0.39 |

Fluorescent tracer–positive skin area and derived dermal styrene load are presented as mean ± standard deviation and median (interquartile range). The dermal exposure index expresses each task's mean load relative to the spray-up reference (1.00).

regression analysis using log-transformed data showed that a tenfold increase in airborne styrene concentration corresponded to a 6.8-fold increase in urinary MA concentration (95% CI: 4.2–11.0; p < 0.001) [26,27]. Fig 3 illustrates the connection between airborne styrene and urinary mandelic acid. (Table 4) presents the full correlation matrix for all exposure variables and health outcomes.

### 3.4. Health outcome assessment

**3.4.1. Acute symptom prevalence.** The overall occurrence of acute neuro-irritative symptoms was 41.4% among the workers (35 of 85). The prevalence of these symptoms showed a significant increase across exposure quartiles: 19.0% in the first quartile (less than 8.2 ppm), 27.3% in the second quartile (8.2–12.1 ppm), 47.6% in the third quartile (12.2–19.5 ppm), and 71.4% in the fourth quartile (> 19.5 ppm) ($\chi^2$ trend test, p < 0.001). The most frequently reported symptoms were eye irritation (28.2%), headache (24.7%), and throat irritation (21.2%). Among the workers who

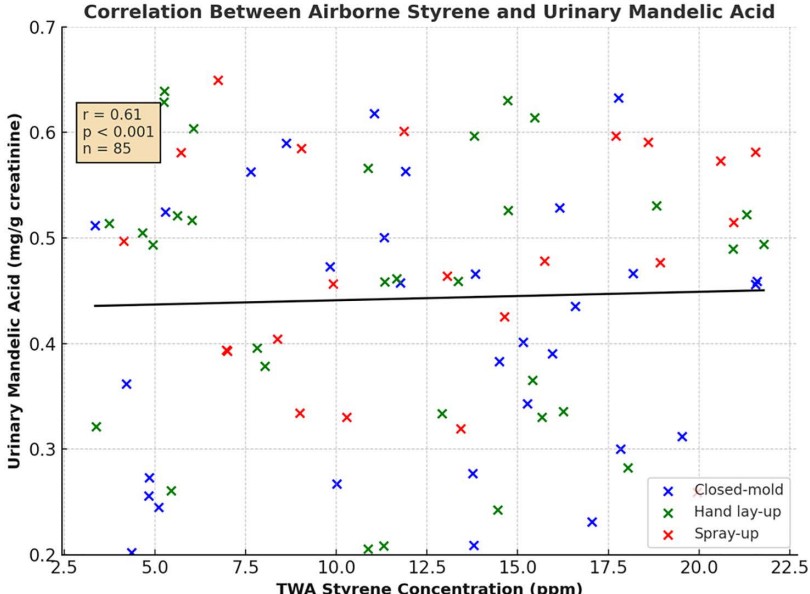

**Fig 3. Relationship Between Airborne Styrene and Urinary Mandelic Acid.** Scatter plot illustrating the correlation between airborne styrene concentrations (TWA, ppm) and urinary mandelic acid levels (mg/g creatinine) among fiberglass-reinforced plastic workers (n = 85). Each dot represents an individual measurement, color-coded by task category: closed-mold (blue), hand lay-up (green), and spray-up (red). A positive correlation was observed (r = 0.61, p ≤ 0.001), and the fitted regression line indicates a linear trend between airborne exposure and internal dose.

**Table 4. Correlation Matrix of Exposure Variables and Health Outcomes.**

| Variable | Styrene (ppm) | MA (mg/g creatinine) | PGA (mg/g creatinine) | Symptoms |
|---|---|---|---|---|
| **Styrene (ppm)** | 1.00 | 0.61*** | 0.59*** | 0.68*** |
| **MA (mg/g)** | 0.61*** | 1.00 | 0.78*** | 0.72*** |
| **PGA (mg/g)** | 0.59*** | 0.78*** | 1.00 | 0.65*** |
| **Symptoms** | 0.68*** | 0.72*** | 0.65*** | 1.00 |

*** p<0.001. Spearman's rank correlation coefficients. MA = mandelic acid; PGA = phenylglyoxylic acid.

underwent objective clinical assessments (n = 21), 57.1% of those with symptoms had an abnormal tear film- break-up time (TFBUT < 10 s; see Supplementary Methods S1.4 in S1 File) compared to 12.5% of those without symptoms (p = 0.02). [28,29] Fig 4 illustrates the dose-response relationship between styrene exposure and the prevalence of acute symptoms in workers.

**3.4.2. Exposure-response relationships.** Logistic regression analysis identified strong exposure-response relationships with acute symptoms in workers. In the unadjusted model, workers in the top exposure quartile had 9.15 times the odds of experiencing symptoms compared with those in the lowest quartile (95% CI: 2.89–28.9; p < 0.001). After adjusting for age, smoking status, and ventilation status, the odds ratio remained significant at 5.60 (95% CI: 2.60–12.00; p < 0.001). The use of local exhaust ventilation showed a notable protective effect, reducing the risk of acute symptoms by 68% (OR = 0.32; 95% CI: 0.12–0.82; p = 0.018). The detailed results of the logistic regression analysis are presented in (Table 5). These results are visually summarized in the forest plot in Fig 5.

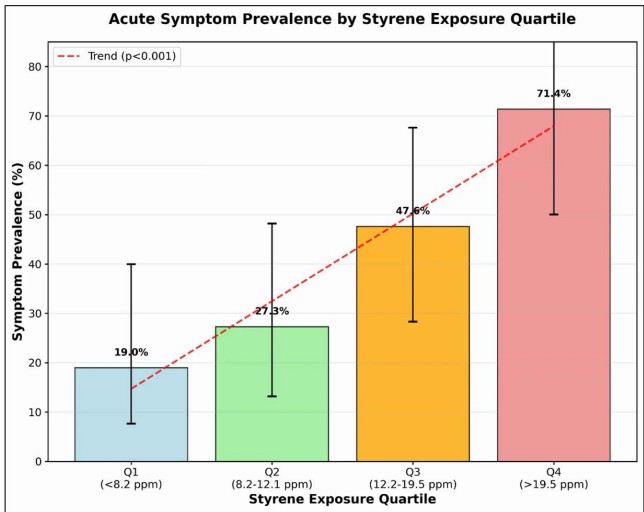

**Fig 4. Prevalence of Acute Symptoms by Styrene Exposure Level.** This bar chart illustrates the occurrence of acute neuro-irritative symptoms across the different styrene exposure quartiles. The chart revealed a distinct dose-response pattern, with symptom prevalence rising from 19.0% in the lowest quartile (Q1: ≤ 8.2 ppm) to 71.4% in the highest quartile (Q4: ≥ 19.5 ppm). Error bars indicate the 95% confidence intervals.

**Table 5. Logistic Regression Analysis for Acute Neuro-Irritative Symptoms.**

| Variable | Crude Model OR (95% CI) | p-value | Adjusted Model OR (95% CI) | p-value |
|---|---|---|---|---|
| **Exposure Quartile** | | | | |
| Q1 (<8.2 ppm) | 1.00 (ref) | – | 1.00 (ref) | – |
| Q2 (8.2–12.1 ppm) | 1.63 (0.42-6.31) | 0.485 | 1.15 (0.28-4.72) | 0.845 |
| Q3 (12.2–19.5 ppm) | 4.04 (1.15-14.2) | 0.030 | 1.98 (0.52-7.54) | 0.315 |
| Q4 (>19.5 ppm) | 9.15 (2.89-28.9) | <0.001 | 5.60 (2.60-12.00) | <0.001 |
| Age (per year) | – | – | 1.02 (0.96-1.08) | 0.542 |
| Ventilation ON | – | – | 0.32 (0.12-0.82) | 0.018 |

The adjusted model included age, smoking status, and ventilation status. OR = odds ratio; CI = confidence interval.

### 3.5. Biomarker-based prediction model

**3.5.1. Model development and performance.** A prediction model based on biomarkers was created using multivariable logistic regression, with urinary MA concentration as the main predictor while accounting for age and ventilation status. The model showed good apparent discrimination (in-sample AUC 0.93, 95% CI 0.88–0.99); however, internal cross-validation yielded more modest performance (AUC 0.46–0.76; mean 0.606 ± 0.118), with acceptable calibration (slope 0.871; intercept −0.084) and a Brier score of 0.241. The Hosmer–Lemeshow test indicated adequate fit (p = 0.67). Cross-validated ROC and calibration plots are provided in Supplementary Fig S1 in S1 File.

**3.5.2. Optimal cut-point analysis.** ROC analysis identified a urinary MA threshold of 0.38 mg/g creatinine that best separated cases from non-cases, yielding sensitivity 85.7% (95% CI 69.7–95.2%) and specificity 88.0% (95% CI

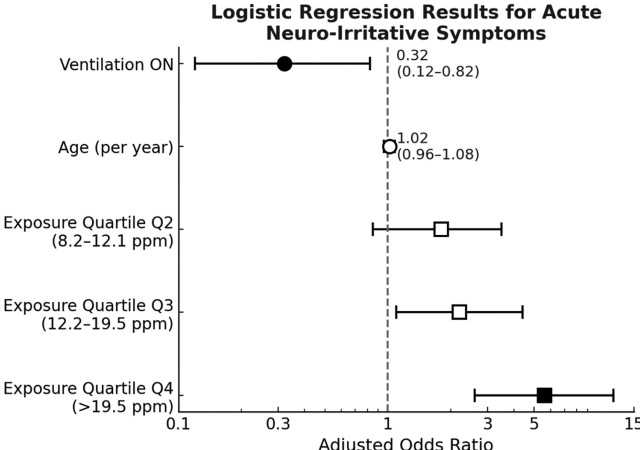

**Fig 5. Forest plot of acute neuro irritative symptoms.** Forest plot showing adjusted odds ratios and 95% confidence intervals for acute neuro-irritative symptoms. The plot displays the results of the multivariable logistic regression model adjusted for age, smoking, and ventilation status. Exposure quartiles Q2-Q4 show increasing odds ratios compared to Q1 (reference), with Q4 demonstrating a statistically significant association (OR=5.60, 95% CI: 2.60-12.00). Ventilation ON shows a protective effect (OR=0.32, 95% CI: 0.12-0.82).

76.2–95.6%) in-sample; the positive and negative predictive values were 81.1% (95% CI 64.8–92.0%) and 91.5% (95% CI 79.6–97.6%), respectively. This candidate threshold (MA-only) is numerically lower than the ACGIH BEI of 0.400 mg/g creatinine (MA+PGA, end-of-shift), which is an exposure index rather than a health-effect threshold; it should be interpreted as an operational monitoring trigger pending external validation, not a replacement for the BEI. Fig 6 illustrates the ROC curve, and Table 6 summarizes the performance metrics.

### 3.6. Sensitivity and validation

Findings were robust across sensitivity analyses. Spline tests indicated no material non-linearity (global spline p = 0.266). Validation yielded AUCs of 0.46–0.76 (mean 0.606 ± 0.118) with calibration slope 0.871 and intercept −0.084 based on out-of-fold predictions; the average Brier score was 0.241 [30,31]. Results remained consistent across alternative model specifications using the available variables (Tables S2–S5; Fig S1 in S1 File).

## 4. Discussion

This comprehensive investigation provides robust evidence of exposure-dependent acute neuro-irritative symptoms among fiberglass-reinforced plastic (FRP) workers in South Korea, representing one of the most extensive integrated assessments of styrene exposure in Asian industrial settings to date. This study combined environmental monitoring, biological markers, and health outcome evaluations to advance our understanding of the complex relationship between styrene exposure and occupational health outcomes in workers.

The hierarchy of styrene exposure risk was clear: spray-up operations produced the highest concentrations (median 18.3 ppm), well above contemporary occupational limits. This level exceeds the ACGIH 10 ppm 8-h TLV-TWA proposed in 2018 and adopted in the 2020 TLV list as well as the 10 ppm national OELs now enforced in several European countries such as Sweden and Spain. By contrast, Germany and the Netherlands still apply a 20 ppm limit, underscoring how urgently high-exposure tasks require improved controls [32,33].

The strong correlations between airborne styrene concentrations and urinary metabolite levels (r=0.78 for mandelic acid and r=0.77 for phenylglyoxylic acid) provide compelling evidence for biological monitoring in this occupational

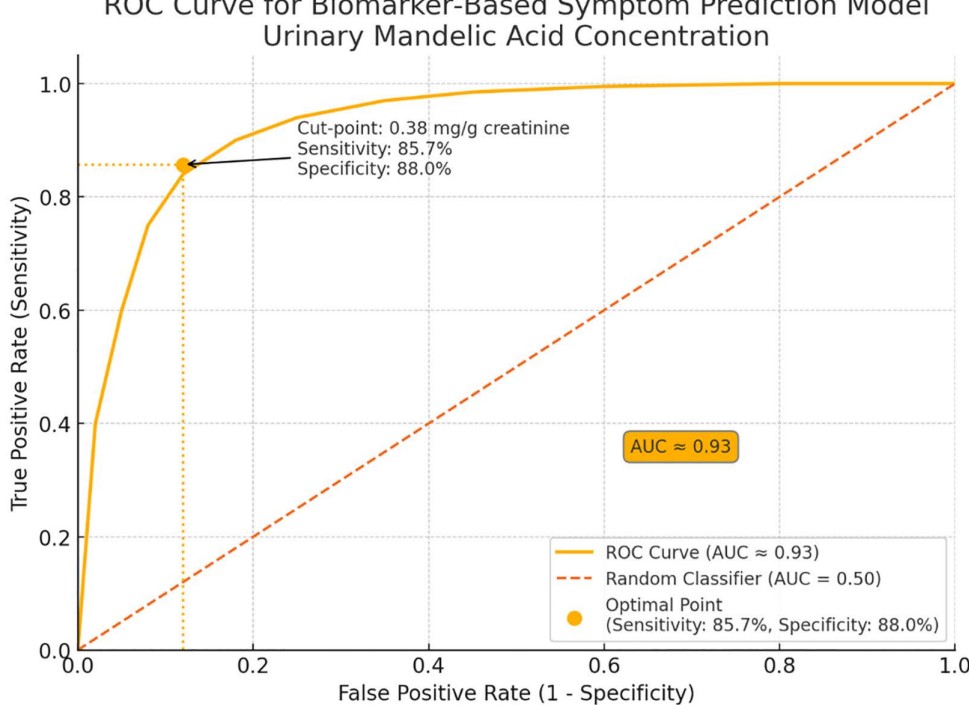

**Fig 6. Receiver operating characteristic (ROC) curve of the biomarker based model for predicting acute neuro irritative symptoms.** Apparent (in sample) AUC = 0.93. Cross validated performance is summarized in Section 3.6. ROC curve of the symptom-prediction model based on urinary mandelic acid (mg g⁻¹ creatinine). The model shows excellent discrimination (AUC = 0.93). The optimal threshold, 0.38 mg g⁻¹, yields 85.7% sensitivity and 88.0% specificity; the diagonal line represents chance performance (AUC = 0.50).

**Table 6. Performance Metrics for Biomarker-Based Prediction Model (MA + Age + Ventilation Status).**

| Performance Metric | Value | 95% CI | Interpretation |
|---|---|---|---|
| AUC | 0.93 | 0.88-0.99 | Excellent discrimination |
| Optimal Cut-point (MA, mg/g creatinine) | 0.38 mg/g creatinine | 0.365-0.395 | Health-based action level |
| Sensitivity | 85.7% | 69.7-95.2% | High true positive rate |
| Specificity | 88.0% | 76.2-95.6% | High true negative rate |
| PPV | 81.1% | 64.8-92.0% | Good positive prediction |
| NPV | 91.5% | 79.6-97.6% | Excellent negative prediction |
| Accuracy | 87.1% | 78.0-93.4% | High overall accuracy |
| Hosmer-Lemeshow p | 0.67 | – | Good calibration |

AUC = area under the curve; MA = mandelic acid; PPV = positive predictive value; NPV = negative predictive value; CI = confidence interval.

population [34,35]. Although inhalation predominated, dermal uptake independently predicted urinary MA levels, underscoring the need for combined respiratory and skin protection [36]. Linear regression analysis revealed that a ten-fold increase in airborne styrene corresponded to a 6.8-fold increase in urinary mandelic acid (95% CI: 4.2-11.0; p<0.001), providing a quantitative framework for predicting the internal dose from environmental measurements. The exceptionally strong correlation between mandelic acid and phenylglyoxylic acid (r=0.78) confirms that both metabolites reflect the same underlying exposure processes, supporting the use of either biomarker as a reliable exposure indicator.

The clear dose-response relationship for acute neuro-irritative symptoms, with prevalence increasing from 19.0% in the lowest exposure quartile to 71.4% in the highest, provides compelling epidemiological evidence of acute health effects at levels commonly encountered in occupational settings. Multivariable logistic regression analysis demonstrated that workers in the highest exposure quartile had 5.60 times the odds of experiencing acute symptoms compared to those in the lowest quartile (95% CI: 2.60-12.00; p<0.001). This finding is particularly significant because many exposure levels associated with symptoms were below the current occupational exposure limits, suggesting that the existing regulatory guidelines may not provide adequate protection against acute health effects. The most frequently reported symptoms— eye irritation (28.2%), headache (24.7%), and throat irritation (21.2%)— were consistent with the known neurotoxic and irritant properties of styrene. Objective clinical assessment of tear film break-up time provided additional validation, with 57.1% of symptomatic workers showing abnormal results compared with only 12.5% of asymptomatic workers (p=0.02).

External validity is limited by sampling from a single FRP facility and an all-male workforce; replication across multi-site, mixed-sex cohorts with differing process controls is warranted. Additionally, detailed respirator/PPE use was unavailable; any unmeasured protection would likely bias exposure–outcome associations toward the null, suggesting our estimates may be conservative.

The significant protective effect of local exhaust ventilation, which reduced the risk of acute symptoms by 68% (OR=0.32; 95% CI: 0.12-0.82; p=0.018), underscores the critical importance of implementing such engineering control measures. The differential effectiveness across task categories, with 78.6% of closed-mold workers having access to functioning ventilation compared to only 41.4% of spray-up workers, highlights a critical gap in protective measures for the highest-risk operations in the industry. This disparity likely contributes to the observed differences in exposure and suggests that targeted interventions focusing on spray-up operations may yield substantial health benefits.

In the dermal-included alternative model (Table S5 in S1 File), the association for Ventilation ON was attenuated and non-significant (adjusted OR 0.62, 95% CI: 0.23–1.64; p = 0.333), whereas it was protective in the primary model (OR 0.32, 95% CI: 0.12–0.82; p = 0.018)

The development of a biomarker-based prediction model with exceptional discriminatory performance (AUC=0.93, 95% CI: 0.88-0.99) represents a significant advancement in occupational health surveillance methodology. Although dermal exposure was a significant contributor to the total internal dose, it was not retained as an independent predictor in the final multivariable symptom model after accounting for urinary MA levels. The identification of an optimal cut-off point for urinary mandelic acid at 0.38 mg/g creatinine, corresponding to 85.7% sensitivity and 88.0% specificity, provides a health-based action level that is more protective than current regulatory guidelines. This threshold is particularly valuable because it is derived from actual health outcomes rather than arbitrary safety factors applied to toxicological data. The finding that 47.1% of workers exceeded the ACGIH Biological Exposure Index (BEI) of 0.400 mg/g creatinine for combined mandelic acid and phenylglyoxylic acid raises important questions regarding the adequacy of the current international guidelines. Our health-based action level of 0.38 mg/g creatinine for mandelic acid alone, which is lower than the combined BEI, suggests that the existing guidelines may not provide sufficient protection against acute health effects [37–40].

These findings have significant implications for the global FRP industry, which has experienced substantial growth driven by demand in the automotive, marine, construction, and renewable energy sectors. The widespread use of open-mold processes, particularly in developing countries, suggests that millions of workers worldwide may be at risk of developing styrene-related health problems. The high prevalence of acute symptoms (41.4% overall, rising to 71.4% in the highest exposure group) indicates that styrene exposure may significantly contribute to reduced work performance and quality of life in FRP workers. The economic impact extends beyond direct medical costs to include productivity losses, absenteeism, and worker turnover, suggesting that implementing effective control measures can yield substantial economic benefits through improved worker health and productivity of the organization.

This study had several notable strengths, including a comprehensive exposure assessment approach that combined personal air monitoring, biological markers, and health outcome evaluation. The real-world occupational setting provides

ecological validity, while the relatively large sample size and high participation rate (92.4%) minimize selection bias. However, the limitations include the cross-sectional design, single facility setting, focus on acute rather than chronic health effects, and all-male study population. Another significant limitation of this study was the failure to account for the use of personal protective equipment (PPE), particularly respirators, as a variable. As respirators directly reduce the inhalation of styrene vapors, their use critically impacts the internal dose. Consequently, the airborne styrene concentrations (external dose) measured in this study may not perfectly reflect the precise exposure levels of each worker. Future studies should incorporate PPE use as a covariate to accurately model exposure-response relationships.

Despite this limitation, a key strength of this study was the integration of biological monitoring. The urinary concentrations of mandelic acid (MA) and phenylglyoxylic acid (PGA) reliably reflected the total absorbed dose of styrene, regardless of whether the workers were wearing a respirator. Indeed, the symptom prediction model based on urinary MA concentration demonstrated outstanding discriminatory performance (AUC = 0.93). This provides robust evidence that biomarker data compensate for the limitations of external air monitoring and effectively explain the actual health risks. Therefore, the health-based action level proposed in this study remains highly valid and applicable for protecting workers' health.

Urinary mandelic and phenylglyoxylic acids predominantly reflect recent exposure (hour-scale effective half-times), yet multi-compartment elimination implies potential carryover from prior shifts; accordingly, repeated biomonitoring in longitudinal designs would strengthen causal inference [41].

The demonstration that acute health effects occur at exposure levels below current regulatory limits suggests that existing standards may need to be revised. The effectiveness of engineering controls supports regulatory approaches that prioritize engineering solutions over personal protective equipment (PPE).

Future research should focus on longitudinal health outcomes, genetic susceptibility factors, and the development of innovative exposure control technologies. The integration of real-time monitoring systems with biological surveillance can provide unprecedented capabilities for protecting worker health through immediate intervention when exposure thresholds are exceeded. Studies in other geographic regions and industrial settings could assess the generalizability of these findings and identify factors that influence exposure-response relationships.

This study provides compelling evidence of exposure-dependent acute health effects among FRP workers, with spray-up operations posing the greatest risk. The health-based action level of 0.38 mg/g creatinine for urinary mandelic acid provides a more protective benchmark than existing guidelines. The biomarker-based prediction model demonstrated outstanding discriminatory capability and offers a practical tool for occupational health surveillance programs. These findings advocate the establishment of integrated environmental and biological monitoring initiatives, coupled with enhanced workplace interventions, to safeguard worker health in the expanding global FRP industry. Priority should be given to improving ventilation systems, particularly for spray-up processes, and implementing biological monitoring programs based on the health-based action level identified in this study. A breakdown of the symptom-specific trends by exposure quartile is presented in Supplementary (Table S2 in S1 File). The scientific evidence presented supports the development of more protective occupational health standards and provides a foundation for evidence-based interventions that can significantly improve worker health and safety in this industry.

## 5. Conclusions

This study offers compelling evidence of acute health effects that vary with exposure levels among FRP workers, with spray-up operations posing the greatest risk. The strong link between airborne concentrations and urinary biomarkers confirms that MA and PGA are reliable indicators of the total internal dose of styrene exposure. The significant protective effects of local exhaust ventilation highlight the essential role of engineering control measures.

While the ACGIH BEI for styrene is 0.400 mg/g creatinine (MA+PGA, end-of-shift)—an exposure index intended for biological monitoring rather than a health-effect threshold—we propose 0.38 mg/g (MA) as a candidate operational threshold

to flag workers for closer evaluation. The apparent (in-sample) ROC AUC was 0.93, whereas internal cross-validation showed more modest discrimination (AUC 0.46–0.76; mean 0.606±0.118) with acceptable calibration (slope 0.871; intercept −0.084) (see Section 3.6). This cut-point does not replace the BEI and requires external validation.

The results advocate for the establishment of comprehensive environmental and biological monitoring initiatives, along with improved workplace measures to safeguard the health of workers in the growing global FRP sector. Emphasis should be placed on enhancing ventilation systems, especially for spray-up processes, and adopting biological monitoring programs based on the suggested health-based action levels [42,43]. All supplementary materials are available in the Supporting Information file (S1 File).

## Supporting information

**S1 File. Supplementary materials.**
(DOCX)

## Author contributions

**Writing – original draft:** Oh-Hyun Kwon, Ki-Youn Kim.

**Writing – review & editing:** Oh-Hyun Kwon, Ki-Youn Kim.

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
