## [Decision Letter · Decision Letter 0]

30 Sep 2025

Dear Dr. Kim,

Thank you for submitting your manuscript to PLOS ONE. After careful consideration, we feel that it has merit but does not fully meet PLOS ONE’s publication criteria as it currently stands. Therefore, we invite you to submit a revised version of the manuscript that addresses the points raised during the review process.

We look forward to receiving your revised manuscript.

Kind regards,

Jin Su Kim, PhD

Academic Editor

PLOS ONE

Journal Requirements:

[No authors have competing interests].

Additional Editor Comments:

Please ensure that the reviewers’ requests are addressed thoroughly

Reviewers' comments:

Reviewer's Responses to Questions

**Comments to the Author**

1. Is the manuscript technically sound, and do the data support the conclusions?

Reviewer #1: Yes

Reviewer #2: Yes

2. Has the statistical analysis been performed appropriately and rigorously?

Reviewer #1: Yes

Reviewer #2: Yes

3. Have the authors made all data underlying the findings in their manuscript fully available?

Reviewer #1: Yes

Reviewer #2: Yes

4. Is the manuscript presented in an intelligible fashion and written in standard English?

Reviewer #1: Yes

Reviewer #2: Yes

Reviewer #1: The study is strong because it combines air monitoring, urine biomarkers, and dermal exposure, giving a complete picture of how workers are exposed. The sample size is good, the methods are solid, and the results clearly show that higher exposure is linked with more symptoms.

Points to improve:

The study is cross-sectional, so you can only say styrene exposure is “associated with” symptoms, not that it directly “causes” them.

The study does not include information on the use of personal protective equipment (like respirators), which is important for interpreting exposure. Please highlight this more clearly as a limitation.

Smoking was considered, but more detail on how it was measured would improve the paper.

The study was done in one factory and only on male workers, so the findings may not apply to all workers. This should be explained in the discussion.

Dermal exposure was measured but not fully included in the final model. Please explain this more clearly.

The proposed new biomarker action level (0.38 mg/g creatinine) is interesting, but explain how it compares to the current ACGIH limit and what it would mean in practice.

Reviewer #2: 1. The paper stated single point measurement of styrene absorption on a narrow range of workers in terms of age, BMI, etc which results in an acute health risk for the workers. But in order to conclude the health impact to the same workers the measurements must be done repeatedly over a period of time. The workers are exposed to styrene before the day of the study and we assume that previous exposure could have affected absorption on the day of the study. this should be considered to make an accurate assessment.

**Do you want your identity to be public for this peer review?** For information about this choice, including consent withdrawal, please see our Privacy Policy

Reviewer #1: No

Reviewer #2: No

---

## [Author Response · Author response to Decision Letter 1]

2 Oct 2025

Dr. Jin Su Kim Academic Editor PLOS ONE

Subject: Resubmission of Revised Manuscript (PONE-D-25-46598)

Dear Dr. Kim,

We are pleased to resubmit the revised version of our manuscript titled, "Occupational Exposure to Styrene and Acute Health Effects among Fiberglass-Reinforced Plastic Workers: An Integrated Environmental and Biological Monitoring Study" (PONE-D-25-46598), for your consideration for publication in PLOS ONE.

We thank you and the reviewers for the insightful and constructive feedback on our initial submission. We have carefully addressed all the points raised by the reviewers and the journal, and we believe the manuscript has been significantly strengthened as a result. A detailed point-by-point response is provided in the 'Response to Reviewers' file.

Key revisions include:

· Clarification of statistical terms (correlation coefficient vs. coefficient of determination) and correction of all related numerical values throughout the manuscript.

· Unification of all data points across the text, tables, and figures to ensure consistency.

· Expansion of the Discussion section to elaborate on the study's limitations (e.g., cross-sectional design, lack of PPE data) and to provide a more in-depth comparison of our proposed biomarker action level with existing guidelines.

· Addition of a new supplementary analysis (Table S5) to further explore the relationship between different exposure routes and health outcomes.

In accordance with the journal's requirements, we have also updated our Data Availability Statement to include a permanent DOI from the Zenodo repository. Our Competing Interests Statement has been updated to read: "The authors have declared that no competing interests exist." Furthermore, the manuscript has been formatted to meet all PLOS ONE style guidelines.

We are confident that the revised manuscript is now suitable for publication in PLOS ONE. We look forward to hearing from you.

Sincerely,

Ki-Youn Kim, PhD Corresponding Author Graduate School of Safety Engineering Seoul National University of Science and Technology

---

## [Editor Report · Decision Letter 1]

6 Oct 2025

Occupational Exposure to Styrene and Acute Health Effects among Fiberglass-Reinforced Plastic Workers: An Integrated Environmental and Biological Monitoring Study

PONE-D-25-46598R1

Dear Dr. %KIM%,

We’re pleased to inform you that your manuscript has been judged scientifically suitable for publication and will be formally accepted for publication once it meets all outstanding technical requirements.

Kind regards,

Jin Su Kim, PhD

Academic Editor

PLOS ONE

---

## [Editor Report · Acceptance letter]

PONE-D-25-46598R1

PLOS ONE

Dear Dr. Kim,

I'm pleased to inform you that your manuscript has been deemed suitable for publication in PLOS ONE. Congratulations! Your manuscript is now being handed over to our production team.

Kind regards,

on behalf of

Dr. Jin Su Kim

Academic Editor

PLOS ONE